# Adaptively Integrative Association between Multivariate Phenotypes and Transcriptomic Data for Complex Diseases

**DOI:** 10.3390/genes14040798

**Published:** 2023-03-26

**Authors:** Yujia Li, Yusi Fang, Hung-Ching Chang, Michael Gorczyca, Peng Liu, George C. Tseng

**Affiliations:** 1Eli Lilly and Company, Indianapolis, IN 46225, USA; 2Department of Biostatistics, University of Pittsburgh, Pittsburgh, PA 15261, USA

**Keywords:** association analysis, gene expression, phenotypes, complex disease

## Abstract

Phenotype–gene association studies can uncover disease mechanisms for translational research. Association with multiple phenotypes or clinical variables in complex diseases has the advantage of increasing statistical power and offering a holistic view. Existing multi-variate association methods mostly focus on SNP-based genetic associations. In this paper, we extend and evaluate two adaptive Fisher’s methods, namely AFp and AFz, from the *p*-value combination perspective for phenotype–mRNA association analysis. The proposed method effectively aggregates heterogeneous phenotype–gene effects, allows association with different data types of phenotypes, and performs the selection of the associated phenotypes. Variability indices of the phenotype–gene effect selection are calculated by bootstrap analysis, and the resulting co-membership matrix identifies gene modules clustered by phenotype–gene effect. Extensive simulations demonstrate the superior performance of AFp compared to existing methods in terms of type I error control, statistical power and biological interpretation. Finally, the method is separately applied to three sets of transcriptomic and clinical datasets from lung disease, breast cancer, and brain aging and generates intriguing biological findings.

## 1. Introduction

Identifying genes associated with disease phenotypes in transcriptomics studies is of long-standing interest. Many methods for testing genes associated with a univariate phenotype have been labelled as “differential expression” analysis, which is named after the commonly encountered case–control design; examples include edgeR [1], limma [2], and DESeq2 [3] (also see review papers [4,5] for more details). In many complex diseases, however, patients are often characterized by multiple phenotypes that reflect different facets of the disease. For example, chronic obstructive pulmonary disease (COPD) can be characterized by FEV1 (i.e., the volume of breath exhaled with effort in one second), FVC (i.e., the full amount of air that can be exhaled with effort in a complete breath), as well as several test indices of blood. Despite active research in univariate association analysis, the problem of testing genes associated with multivariate phenotypes is understudied and presents new challenges.

A naive association is to test the association between all pairs of genes and phenotypes and interpret significant genes for each phenotype. This, however, loses the statistical power and interpretation of gene mechanisms in often correlated phenotypes. Therefore, there is great interest in aggregating the effects of a gene on multiple phenotypes and testing for the overall association. Hence in this paper, we consider the following union–intersection test (UIT) [6] or conjunction null hypothesis [7] to determine the overall association between a gene and *K* phenotypes of interest:$$\begin{array}{c} H_{0}:\vec{\theta }\in ⋂\left\{\theta_{k}=0\right\}\\ H_{A}:\vec{\theta }\in ⋃\left\{\theta_{k}\neq 0\right\}. \end{array}$$
where $\vec{\theta }=\left(\theta_{1},\ldots ,\theta_{K}\right)$ and each $\theta_{k}$ denotes the effect size for measuring the association between the gene and phenotype *k*. Conventional methods for the UIT test include multivariate analysis of variance (MANOVA), linear mixed models (LMMs), and generalized linear models (GLMM). However, such methods either impose strong assumptions on the data type of phenotypes (e.g., Gaussian assumption for MANOVA) or are incapable of handling multiple phenotypes of different data types (e.g., continuous, binary and categorical) at the same time. As it is common to have phenotypes of various data types in a complex disease, it is challenging to use the above methods to aggregate the effects of multiple phenotypes in practice. Another category of potential solution is regression-based methods. O’Reilly et al. [8] proposed the MultiPhen method by regressing genotypes (SNPs in the paper but genes in our case) on phenotypes via a proportional odds logistic model, which is only applicable to ordinal outcomes in GWAS. Wu et al. [9] proposed the multi-trait gene sequence kernel association test (MSKAT), where the purpose is to test the association between a phenotype and multiple SNPs in a chromesome region. Since both regression-based methods were previously developed for SNP-based association and are difficult to extend to the transcriptomic scenario, we will not include them in the evaluation in this paper.

Borrowing ideas from meta-analysis, another approach is to combine summary statistics (e.g., *p*-values and test statistics) from the association tests that separately test for each phenotype. O’Brien et al. [10] combined test statistics from the individual tests on each trait weighted by inverse variance. Pan et al. [11] and Zhang et al. [12] proposed the sum of powered score tests (SPU) and adaptive SPU (aSPU) to combine the score test statistics derived from generalized estimation equations (GEE). However, the proposed methods are still incapable of dealing with phenotypes of different data types. In addition, the two methods involve tuning parameters and are not fully data-driven. Instead of directly combining test statistics, many *p*-value combination tests are also applicable to this problem. Examples include Fisher’s method [13] ($T^{\mathrm{Fisher}}=-2\sum_{k=1}^{K}\log p_{k}$), or the minimum *p*-value method (ref. [14] denoted by minP hereafter) ($T^{\mathrm{minP}}=\min p_{i}$), where for each k=1,…,K, $p_{k}$ denotes the *p*-value of hypothesis test that tests the association between the gene and phenotype *k*. To account for dependency between phenotypes and thus the *p*-values, one can perform a permutation test by randomly shuffling sample columns in the gene expression data while keeping the phenotype data unchanged. In this way, the associations between genes and phenotypes break down while correlation structure between phenotypes is preserved. Van der Sluis et al. [15] also proposed to extend the Simes’ test by exploiting the correlation structure between *p*-values (TATES). The *p*-value combination approaches can aggregate effects from multiple phenotypes of arbitrary data types, as long as the input *p*-values are derived from valid tests.

In addition to association tests, another challenge arises from heterogeneity across phenotypes, where only a subset of phenotypes have associations and the subsets differ for each gene. That is, in a multivariate phenotype–gene association study on a complex disease, a given gene may only have an association with a subset of phenotypes, and the association levels may be heterogeneous. The heterogeneity of associations can significantly impact the power of existing methods. For example, when the subset of phenotypes associated with the gene is small or there is only a single strong effect with many rather weak effects, the minP method will be more powerful, while when most phenotypes have associations, Fisher’s method will perform better [16]. The heterogeneity can also lead to a misleading biological interpretation of existing methods. For example, suppose ${\vec{p}}_{1}=\left(0.001,1,1\right)$ denotes *p*-values for separately testing associations between gene 1 and three phenotypes, and ${\vec{p}}_{2}=\left(0.1,0.1,0.1\right)$ denotes the *p*-values for gene 2. Although both genes lead to the same Fisher’s test statistics ($T^{\mathrm{Fisher}}=13.8$), their biological interpretations hugely differ. ${\vec{p}}_{1}$ indicates strong statistical significance in only the first phenotype, whereas ${\vec{p}}_{2}$ indicates marginal statistical significance in all three. All the methods mentioned above fail to characterize the heterogeneity across the phenotypes. As a result, methods for aggregating heterogeneous effects that have a refined characterization of heterogeneity and competitive statistical power are preferred for this problem.

Another challenge is to identify the subset of phenotypes that are associated with a given gene. When the individual associations between the phenotypes and the gene are strong, it is easy to look at whether each corresponding *p*-value is significant or not. However, in our setting, where each individual association between the phenotype and the gene is weak (corresponding *p*-value is marginally significant or non-significant), it is challenging to identify the subset of associated phenotypes. To the best of our knowledge, no existing method has addressed this problem.

To address the above challenges, we extend two existing *p*-value combination methods to aggregate association information across phenotypes. In Li and Tseng [17] and Huo et al. [18], an adaptively weighted Fisher’s statistic takes the minimal *p*-values among all possible combinatorial aggregations. Consequently, we denote the procedure modified from this method as AFp. Later, Song et al. [19] proposed an alternative adaptive Fisher’s statistics by taking the minimal *z*-standardized Fisher score (i.e., *z*-standardization by subtraction of the mean and division of standard deviation) among all possible combinatorial aggregations, which we consequently denote as its modified method as AFz in this paper. Based on the *p*-value combination, both methods can accommodate multiple phenotypes of different data types at the same time. Furthermore, both methods adaptively assign a binary 0–1 weight to each input *p*-value to determine whether the corresponding phenotype is associated with the gene of interest. As a result, the proposed methods efficiently aggregate heterogeneous effects across phenotypes, provide a refined characterization of the heterogeneity of association patterns for better biological interpretation, and identify the subset of associated phenotypes, even when the individual associations are weak. We evaluate the two methods through comprehensive simulations. We find that both methods have comparable performance on type I error control and statistical power for detecting the overall association between a gene and multiple phenotypes. However, the AFp-based method outperforms the AFz-based method in phenotype selection. Consequently, we recommend the AFp-based method for future applications. For further downstream interpretation, we propose to evaluate the variability of the 0–1 weight estimates using variability indices constructed through bootstrapping. In addition, following Huo et al. [18], we further use the bootstrap samples to estimate a co-membership matrix of genes, followed by the tight clustering method [20] to identify gene modules of different phenotype association patterns.

The paper is structured as follows. Section 2 introduces the AFp and AFz-based methods and the accompanying downstream analysis methods. Section 3.1 performs extensive simulations to evaluate AFp, AFz, as well as other existing methods. Section 3.2 contains the results from a lung disease transcriptomic dataset, a breast cancer transcriptomic dataset, and a brain aging transcriptomic dataset, respectively. Section 4 includes final conclusion and discussion.

## 2. Methods

This section introduces our proposed methods for multivariate phenotype-gene association analysis. Section 2.1 discusses how to generate the input *p*-values. In Section 2.2, we propose to use the adaptive sum of log-transformed *p*-values to aggregate heterogeneous effects in the input *p*-values, which is the foundation that AFz and AFp are built upon. We also propose a permutation procedure to take the correlation structure of phenotypes into account. By extending the AFp and AFz methods, Section 2.3 and Section 2.4 propose two ways to adaptively assign the binary 0–1 weights for the log-transformed *p*-values and formulate the final statistic for testing the overall association between multiple phenotypes and the gene of interest. Section 2.5 and Section 2.6 discuss the bootstrap algorithms to estimate the variability indices of the 0–1 weight estimates and identify gene modules associated with disease phenotypes. Suppose there are *n* independent samples, *K* phenotypes, *p* gene features, and *M* covariates. Denote by $Y_{ik}$, $Z_{im}$, and $X_{ij}$ the *k*-th phenotype, *m*-th covariate, and *j*-th gene feature of subject *i*, respectively, where 1≤k≤K, 1≤m≤M, and 1≤j≤p. Let $\vec{Y_{k}}={\left(Y_{1k},Y_{2k},\ldots ,Y_{nk}\right)}^{T}$, $\vec{Z_{m}}={\left(Z_{1m},Z_{2m},\ldots ,Z_{nm}\right)}^{T}$, and $\vec{X_{j}}={\left(X_{1j},X_{2j},\ldots ,X_{nj}\right)}^{T}$ be the vectors of the *k*-th phenotype, *m*-th covariate, and *j*-th gene for all the samples, respectively.

### 2.1. Generalized Linear Models for Generating Input p-Values

To generate the input *p*-values for both the AFp and AFz methods, the following generalized linear model is assumed for the *k*-th phenotype and the *j*-th gene with *M* covariates:(1)$$g_{k}\left(E\left(Y_{ik}\right)\right)=X_{ij}\cdot \theta_{jk}+\sum_{m=1}^{M}Z_{im}\cdot \alpha_{mkj}$$
where $\theta_{jk}$ is the coefficient of the gene feature, $\alpha_{mkj}$ is the coefficient of covariates, and $g_{k}\left(\right)$ is the link function. The associated *p*-value $p_{jk}$ of $\theta_{jk}$ can be derived by the classic Wald test or score test, which serves as the input *p*-values for the combination. Note that the distributional assumptions on $Y_{ik}$ and the choices of link function $g_{k}$ are not required to be consistent across the phenotypes, which allows one to accommodate phenotypes of different data types. For example, one can use $g_{k}\left(t\right)=\mathrm{logit}\left(t\right)$ for binary outcomes, $g_{k}\left(t\right)=\log\left(t\right)$ for count data, and $g_{k}\left(t\right)=t$ for continuous outcomes.

### 2.2. Aggregation of Heterogeneous and Dependent Effects in p-Values

To aggregate heterogeneous effects in the input *p*-values, we propose to use the following weighted sum of the log-transformed *p*-values:$$U_{j}\left(\vec{w_{j}}\right)=-\sum_{k=1}^{K}w_{jk}\log\left(p_{jk}\right),$$
where $w_{jk}\in \left\{0,1\right\}$ is the binary 0–1 weight to determine whether the *k*-th phenotype is associated with the *j*-th gene. Here, $\vec{w_{j}}={\left(w_{j1},w_{j2},\ldots ,w_{jK}\right)}^{T}$ denotes the weight vector. The observed weighted statistic of $U_{j}\left(\vec{w_{j}}\right)$ is denoted by $u_{j}\left(\vec{w_{j}}\right)$. Let $\Omega =\left\{\vec{w_{j}}:\vec{w_{j}}=\left(w_{j1},\ldots ,w_{jK}\right)\in {\left\{0,1\right\}}^{K}\right\}\backslash \left\{\vec{0}\right\}$ be the searching space of all possible realizations of $\vec{w_{j}}$. In the following Section 2.3 and Section 2.4, we adapt the AFp and AFp methods to aggregate $U_{j}\left(\vec{w_{j}}\right)$s with all possible choices of $w_{j}$ in Ω and adaptively choose the best $\vec{w_{j}}$ to determine the subset of *p*-values with true signals. The above procedures require estimation of the mean, standard deviation, and *p*-value of $U_{j}\left(\vec{w_{j}}\right)$ for a given $\vec{w_{j}}$. However, those quantities are intractable when a complex dependency structure between the phenotypes is present. To account for the correlations between the phenotypes, we propose the following permutation procedure.

Recall that we want to test the conditional independence between each gene and *K* phenotypes given the covariates Z (i.e., $Y⊥\vec{X_{j}}|Z$), where $Z=(\vec{Z_{1}},\vec{Z_{2}},\ldots \vec{Z_{m}})$ and $Y=(\vec{Y_{1}},\vec{Y_{2}},\ldots \vec{Y_{K}})$. We intend to develop a permutation procedure that breaks the associations between Y and $\vec{X_{j}}$ while preserving the associations between Y and Z and between $\vec{X_{j}}$ and Z. Simply permuting the genotype $\vec{X_{j}}$ leads to an inflated type I error rate because it also breaks the correlations between $\vec{X_{j}}$ and covariates Z. Following Potter [21] and Werft and Benner [22], we permute residuals of regressions of $\vec{X_{j}}$ on Z for generalized regression models. That is, we first regress each gene feature on the covariates, then permute the residuals derived from the regression and fit the generalized linear model by regressing each phenotype on the permuted residuals.

More precisely, we denote the vector of residuals of regressing $X_{j}$ on Z by $\vec{e_{j}}={\left(e_{1j},e_{2j},\ldots ,e_{nj}\right)}^{T}$ and permute it for *B* times. In the *b*-th permutation, we regress $\vec{Y_{k}}$ on $e_{j}^{\left(b\right)}$ by the generalized linear model $g_{k}\left(E\left(Y_{ik}\right)\right)=e_{ij}^{\left(b\right)}\cdot \theta_{jk}^{\left(b\right)}$ (1≤i≤n) and calculate the *p*-values $p_{jk}^{\left(b\right)}$ for the coefficient $\theta_{jk}^{\left(b\right)}$. After *B* permutations, we obtain a B×K matrix $P=\{p_{jk}^{\left(b\right)}\}$. The observed weighted statistics for permuted data are calculated as $u_{j}^{\left(b\right)}\left(\vec{w_{j}}\right)=-\sum_{k=1}^{K}w_{jk}\log\left(p_{jk}^{\left(b\right)}\right)$. Note that we break the association of each gene and each phenotype. Therefore, for a given $\vec{w_{j}}$, the null distribution of the weighted statistic $U_{j}\left(\vec{w_{j}}\right)$ is approximated by $\left\{u_{j^{\prime }}^{\left(b\right)}\left(\vec{w_{j}}\right),1\leq b\leq B,1\leq j^{\prime }\leq p\right\}$ with precision B×p.

### 2.3. AFp

Below, we extend the adaptively weighted Fisher’s method [17,18] to aggregate *p*-values from each dependent phenotype. Since the statistics take the minimal *p*-values among all possible combinatorial aggregation, we denote the method as AFp. Under the null hypothesis that $\theta_{jk}=0$, ∀k, the *p*-value of observed weighted statistic, $p_{U}\left(u_{j}\left(\vec{w_{j}}\right)\right)$, can be obtained for the *j*-th gene and a given weight $\vec{w_{j}}\in \Omega$ by
$$p_{U}\left(u_{j}\left(\vec{w_{j}}\right)\right)=\frac{\sum_{b=1}^{B}\sum_{j^{\prime }=1}^{p}I\left\{u_{j^{\prime }}^{\left(b\right)}\left(\vec{w_{j}}\right)\geq u_{j}\left(\vec{w_{j}}\right)\right\}}{B\cdot p}.$$

Here, I{.} is the indicator function. We have further defined the AFp statistic as the minimal *p*-value among all possible weights $\vec{w_{j}}\in \Omega$ (see Li and Tseng [17] and Huo et al. [18] for more details):$$T_{j}^{\mathrm{AFp}}=\min_{\vec{w_{j}}\in \Omega }p_{U}\left(u_{j}\left(\vec{w_{j}}\right)\right).$$

The estimated weight vector
$${\hat{\vec{w_{j}}}}^{AFp}=\arg\min_{\vec{w_{j}}\in \Omega }p_{U}\left(u_{j}\left(\vec{w_{j}}\right)\right)$$
can be used to determine the subset of phenotypes associated with the gene *j*. To calculate the *p*-value of $T_{j}^{\mathrm{AFp}}$, we similarly calculate $T_{j}^{\mathrm{AFp},(b)}$ using $P=\{p_{jk}^{\left(b\right)}\}$ from permutation. Specifically, we calculate
$$T_{j}^{\mathrm{AFp},(b)}=\min_{\vec{w_{j}}\in \Omega }p_{U}\left(u_{j}^{\left(b\right)}\left(\vec{w_{j}}\right)\right),$$
where $p_{U}\left(u_{j}^{\left(b\right)}\left(\vec{w_{j}}\right)\right)=\frac{\sum_{b=1}^{B}\sum_{j^{\prime }=1}^{p}I\left\{u_{j^{\prime }}^{\left(b\right)}\left(\vec{w_{j}}\right)\geq u_{j}^{\left(b\right)}\left(\vec{w_{j}}\right)\right\}}{B\cdot p}$, and the *p*-value of $T_{j}^{\mathrm{AFp}}$ can be calculated as
$$p_{T}\left(T_{j}^{AFp}\right)=\frac{\sum_{b=1}^{B}\sum_{j^{\prime }=1}^{p}I\left\{T_{j^{\prime }}^{AFp,(b)}\leq T_{j}^{AFp}\right\}}{B\cdot p}.$$

In summary, $p_{T}\left(T_{j}^{AFp}\right)$ can be used to determine whether the *j*-th gene is associated with *K* phenotypes and ${\hat{\vec{w_{j}}}}^{AFp}$ can be used to determine which specific phenotypes the *j*-th gene is associated with.

### 2.4. AFz

Below, we extend another adaptive Fisher’s method [19] to aggregate *p*-values from each dependent phenotype. Since the statistics take the minimal standardized Fisher’s score (i.e., *z*-standardization by subtracting the mean and dividing by the standard deviation) among all possible combinatorial aggregation, we denote the method as AFz. For a given $\vec{w_{j}}\in \Omega$, the mean and standard deviation of $u_{j}\left(\vec{w_{j}}\right)$ under null can be approximated by $\hat{E}\left(u_{j}\left(\vec{w_{j}}\right)\right)=\frac{\sum_{b=1}^{B}\sum_{j^{\prime }=1}^{p}u_{j^{\prime }}^{\left(b\right)}\left(\vec{w_{j}}\right)}{B\cdot p}$ and $\hat{sd}\left(u_{j}\left(\vec{w_{j}}\right)\right)=\sqrt{\frac{\sum_{b=1}^{B}\sum_{j^{\prime }=1}^{p}{\left\{u_{j^{\prime }}^{\left(b\right)}\left(\vec{w_{j}}\right)-E\left(u_{j}\left(\vec{w_{j}}\right)\right)\right\}}^{2}}{B\cdot p}},$ respectively. We denote by $u_{j}^{\prime }\left(\vec{w_{j}}\right)$ the *z*-standardized (observed) weighted statistic (see [19]),
$$u_{j}^{\prime }\left(\vec{w_{j}}\right)=\frac{u_{j}\left(\vec{w_{j}}\right)-\hat{E}\left(u_{j}\left(\vec{w_{j}}\right)\right)}{\hat{sd}\left(u_{j}\left(\vec{w_{j}}\right)\right)},$$
and the AFz statistic is defined as the largest standarized observed weighted statistic among all possible weights:$$T_{j}^{\mathrm{AFz}}=\max_{\vec{w_{j}}\in \Omega }u_{j}^{\prime }\left(\vec{w_{j}}\right).$$

The estimated weight vector is obtained by
$${\hat{\vec{w_{j}}}}^{AFz}=\arg\max_{\vec{w_{j}}\in \Omega }u_{j}^{\prime }\left(\vec{w_{j}}\right).$$

To calculate the *p*-value of $T_{j}^{\mathrm{AFz}}$, we obtain the standardized observed weighted statistic by ${u_{j}^{\prime }}^{,(b)}\left(\vec{w_{j}}\right)=\frac{u_{j}^{\left(b\right)}\left(\vec{w_{j}}\right)-\hat{E}\left(u_{j}^{\left(b\right)}\left(\vec{w_{j}}\right)\right)}{\hat{sd}\left(u_{j}^{\left(b\right)}\left(\vec{w_{j}}\right)\right)}$ for each permutation *b*, and $T_{j}^{\mathrm{AFz},(b)}=\max_{\vec{w_{j}}\in \Omega }{u_{j}^{\prime }}^{,(b)}\left(\vec{w_{j}}\right)$, where $\hat{E}\left(u_{j}^{\left(b\right)}\left(\vec{w_{j}}\right)\right)=\hat{E}\left(u_{j}\left(\vec{w_{j}}\right)\right)$ and $\hat{sd}\left(u_{j}^{\left(b\right)}\left(\vec{w_{j}}\right)\right)=\hat{sd}\left(u_{j}\left(\vec{w_{j}}\right)\right)$ by definition. Finally, the *p*-value of ${\vec{w_{j}}}^{AFz}$ is calculated as
$$p_{T}\left(T_{j}^{AFz}\right)=\frac{\sum_{b=1}^{B}\sum_{j^{\prime }=1}^{p}I\left\{T_{j^{\prime }}^{AFz,(b)}\geq T_{j}^{AFz}\right\}}{B\cdot p}$$

In summary, $p_{T}\left(T_{j}^{AFz}\right)$ can be used to determine whether the *j*-th gene is associated with *K* phenotypes, and ${\hat{\vec{w_{j}}}}^{AFz}$ can be used to determine which specific phenotypes the *j*-th gene is associated with.

One can note that both AFp and AFz use either the minimum *p*-value (AFp) or maximum *z*-score (AFz) as the final test statistic, where each *p*-value (AFp) or *z*-score (AFz) corresponds to a subset of selected phenotypes (phenotypes with weights equal to 1), and all possible subsets are searched in the optimization. The rationale comes from the fact that the traditional Fisher’s method combines all *p*-values, including non-signal ones. This greatly weakens statistical power. Both AFp and AFz aim to identify the most possible subset of phenotypes that are associated with the gene by an optimization procedure in the test statistics. This strategy of adaptive combination of *p*-values has gradually become popular in the literature, e.g., [17,19,23,24,25,26].

### 2.5. Variability Index of Adaptive Weights

The 0–1 weight estimates ${\hat{w}}_{j}=\left({\hat{w}}_{j1},\ldots ,{\hat{w}}_{jK}\right)$ obtained by either AFp or AFz are binary and discontinuous as a function of the input *p*-values and thus may not be stable. Following [18], we use a bootstrap procedure to calculate an estimate of variability index $U_{jk}=4\cdot \mathrm{Var}\left({\hat{w}}_{jk}\right)$ for the *j*-th gene and *k*-th phenotype, where the normalization factor 4 scales ${\hat{w}}_{jk}$ to [0,1]. We obtain *L* bootstrap samples with ${\vec{Y_{k}}}^{\left(l\right)}$, ${\vec{Z_{m}}}^{\left(l\right)}$ and ${\vec{X_{j}}}^{\left(l\right)}$ for the *k*-th phenotype, *m*-th covariate, and *j*-th gene, where 1≤k≤K, 1≤m≤M, 1≤j≤G, *l* is the bootstraping index and 1≤l≤L. Following the same procedure in Section 2.3 and Section 2.4, weight estimates for AFp and AFz can be estimated as ${\hat{\vec{w_{j}}}}^{AFp,(l)}=\left({\hat{w}}_{j1}^{AFp,(l)},\ldots ,{\hat{w}}_{jK}^{AFp,(l)}\right)$ and ${\hat{\vec{w_{j}}}}^{AFz,(l)}=\left({\hat{w}}_{j1}^{AFz,(l)},\ldots ,{\hat{w}}_{jK}^{AFz,(l)}\right)$ for the *l*-th bootstrap and *j*-th gene. The final variability indices are obtained by
$${\hat{U}}_{jk}^{AFp}=\frac{4}{L}\sum_{l=1}^{L}\left({\hat{w}}_{jk}^{AFp,(l)}-\frac{1}{L}\sum_{l^{\prime }=1}^{L}{\hat{w}}_{jk}^{AFp,(l)}\right){}^{2}$$
and
$${\hat{U}}_{jk}^{AFz}=\frac{4}{L}\sum_{l=1}^{L}\left({\hat{w}}_{jk}^{AFz,(l)}-\frac{1}{L}\sum_{l^{\prime }=1}^{L}{\hat{w}}_{jk}^{AFz,(l)}\right){}^{2},$$
respectively.

### 2.6. Ensemble Clustering for Biomarker Categorization

As mentioned in the end of Section 2.4, a unique advantage of AFp and AFz is to estimate the 0–1 weights to identify the subset of associated phenotypes for a given gene (a weight of 1 or 0 means the phenotype is associated or independent with the gene, respectively). Consequently, the methods optimize and select from all possible subsets (i.e., $2^{K}-1$ combinations of the 0–1 weight values), which grow exponentially. When further considering the sign (positive/negative) of the association in each phenotype, the number of possible association patterns increase to $3^{K}-1$. For example, in the lung disease application (Figure 1), five phenotypes generate a total of $3^{5}-1=242$ possible phenotype association patterns. To overcome this challenge, we performed a gene clustering procedure proposed by [18] to identify data-driven gene modules ($M_{1},M_{2},\cdots M_{q}$) of major phenotype association patterns. We clustered genes by a co-membership matrix for all pairs of genes where each element of the co-membership matrix represents a similarity of signed weight $\hat{v}=\hat{w}\times$ sign($\hat{\theta }$) of any pair genes. Similar to Section 2.5, we bootstrapped data *L* times and obtained the signed weight statistics ${\hat{v_{jk}}}^{AFp,(l)}={\hat{w_{jk}}}^{AFp,(l)}\times \mathrm{sign}\left({\hat{\theta_{jk}}}^{AFp,(l)}\right)$ and ${\hat{v_{jk}}}^{AFz,(l)}={\hat{w_{jk}}}^{AFz,(l)}\times \mathrm{sign}\left({\hat{\theta_{jk}}}^{AFz,(l)}\right)$ for the *j*-th gene, *k*-th phenotype, and *l*-th bootstrapping data for AFp and AFz, respectively. We next calculated the co-membership matrix for *l*-th bootstrapping data of AFp as $V^{AFp,(l)}\in R^{p\times p}$, where $V_{jj^{\prime }}^{AFp,(l)}=1$ if ${\hat{v}}_{jk}{}^{AFp,(l)}={\hat{v}}_{j^{\prime }k}{}^{AFp,(l)}$ for all *k*, and $V_{jj^{\prime }}^{AFp,(l)}=0$ otherwise. The final co-membership matrix was calculated as $V^{AFp}=\sum_{l=1}^{L}V^{AFp,(l)}/L$ and any classic clustering algorithm could be applied to obtain gene categorization. In this paper, we applied the tight clustering method [20] in real applications, which can eliminate the distractions of scattered genes and construct compact gene modules. $V^{AFz}$ can also be obtained in a similar way, followed by the tight clustering algorithm.

## 3. Results

### 3.1. Simulation

In this section, we perform three simulations to evaluate: (1) Type I error control and power for AFp and AFz-based methods and other existing methods; (2) the accuracy of the weight estimates of the AFp and AFz-based methods. The methods evaluated include MANOVA, aSPU.ind, aSPU.ex, TATES, Fisher, minP, AFp, and AFz. In Simulations I and II, we consider continuous phenotypes without and with confounders, respectively, and all the methods above are evaluated. In Simulation III, we consider a mixture of phenotypes of count and continuous types, and we benchmark the performance of TATES, Fisher, minP, AFp, and AFz since other methods are not applicable. We have two different settings, A and B, in each of Simulations I, II, and III, where A mimics the scenarios where each phenotype-gene association has a similar effect size, and B generates the scenarios where some phenotypes have much stronger associations with genes compared to other phenotypes. In each simulation setting, we adapt a random effect model to simulate a hierarchical association structure between 10 phenotypes and 4800 genes, where phenotypes 1 to 4 are associated with genes 1 to 1600, phenotypes 5 to 9 are associated with genes 1601 to 3200, and phenotype 10 is associated with genes 3201 to 4800. The details of each simulation setting are illustrated below.

#### 3.1.1. Simulation Settings

Simulation IA and IB:

Simulation I simulated continuous phenotypes without confounders.

We simulated $u_{i1},u_{i2},u_{i3}∼N\left(0,\sigma_{\mu }^{2}\right)$ for each sample, where N() stands for a Gaussian distribution and $1\leq i\leq N_{1}$. $N_{1}$ is the sample size.We simulated 10 phenotypes, where $y_{ik}∼N\left(u_{i1},\sigma_{k}^{2}\right)$ for 1≤k≤4, $y_{ik}∼N\left(u_{i1}+u_{i2},\sigma_{k}^{2}\right)$ for 5≤k≤9 and $y_{i10}∼N\left(u_{i3},\sigma_{10}^{2}\right)$.We simulated 4800 gene features, where $x_{ij}∼N\left(u_{i1},\sigma_{x}^{2}\right)$ for 1≤j≤1600, $x_{ij}∼N\left(u_{i2},\sigma_{x}^{2}\right)$ for 1601≤j≤3200 and $x_{ij}∼N\left(u_{i3},\sigma_{x}^{2}\right)$ for 3201≤j≤4800.

We set $N_{1}\in \left\{100,30\right\}$, $\sigma_{x}=0.5$, and $\sigma_{\mu }\in \left\{0,0.4,0.6\right\}$ for varying correlation levels. When $\sigma_{\mu }=0$, all the phenotypes are independent of genes, and the larger the $\sigma_{\mu }$ is, the stronger association between phenotypes and genes. For $\sigma_{k}$, we chose two values that corresponded to two different scenarios, respectively. In Simulation IA, we set $\sigma_{k}=2$ for 1≤k≤9 and $\sigma_{10}=1$, where each phenotype–gene association has a similar effect size. In Simulation IB, we set $\sigma_{1}=\sigma_{5}=0.05$, $\sigma_{10}=1$, and $\sigma_{k}=2$ otherwise, where $\sigma_{1}=\sigma_{5}=0.05$ ensures that the first phenotype has much significant association with genes 1 to 1600 compared with phenotypes 2 to 9 and the 5-th phenotype has much significant association with genes 1601 to 3200 compared with phenotypes 5 to 9. We used Simulation IB to evaluate the performance when some phenotypes have a much stronger association with genes compared with other phenotypes.

Simulation IIA and IIB:

Simulation II simulated continuous phenotypes with a confounder *z* for genes 1 to 1600 and phenotypes 1 to 9.

We simulated $u_{i1},u_{i2},u_{i3}∼N\left(0,\sigma_{\mu }^{2}\right)$ and $z_{i}∼N\left(0,\sigma_{c}^{2}\right)$ where N() stands for a Gaussian distribution and $1\leq i\leq N_{1}$. $N_{1}$ is the sample size.We simulated 10 phenotypes, where $y_{ik}∼N\left(u_{i1}+z_{i},\sigma_{k}^{2}\right)$ for 1≤k≤4, $y_{ik}∼N\left(u_{i1}+u_{i2}+z_{i},\sigma_{k}^{2}\right)$ for 5≤k≤9, and $y_{i10}∼N\left(u_{i3},\sigma_{10}^{2}\right)$.We simulated gene expression data for 4800 genes, where $x_{ij}∼N\left(u_{i1}+z_{i},\sigma_{x}^{2}\right)$ for 1≤j≤1600, $x_{ij}∼N\left(u_{i2},\sigma_{x}^{2}\right)$ for 1601≤j≤3200, and $x_{ij}∼N\left(u_{i3},\sigma_{x}^{2}\right)$ for 3201≤j≤4800.

Similar to Simulation I, we set $N_{1}\in \left\{100,30\right\}$, $\sigma_{x}=0.5$, and $\sigma_{\mu }\in \left\{0,0.4,0.6\right\}$. In Simulation IIA, we set $\sigma_{k}=2$ for 1≤k≤9, and $\sigma_{10}=1$, and in Simulation IIB, we set $\sigma_{1}=\sigma_{5}=0.05$, $\sigma_{10}=1$, and $\sigma_{k}=2$ otherwise.

Simulation IIIA and IIIB:

Simulation III generated phenotypes with a mixture of count, binary, and continuous-type data.

We simulated $u_{i1},u_{i2},u_{i3}∼N\left(0,\sigma_{\mu }^{2}\right)$ where N() stands for a Gaussian distribution and $1\leq i\leq N_{1}$. $N_{1}$ is the sample size.We simulated 10 phenotypes, where $y_{ik}∼Possion\left(u_{i1}\right)$ for 1≤k≤4, $y_{i5}∼Bin\left(1,\frac{\exp(u_{i1}+u_{i2})}{1+\exp(u_{i1}+u_{i2})}\right)$, $y_{ik},∼N\left(u_{i1}+u_{i2},\sigma_{k}^{2}\right)$ for 6≤k≤9, and $y_{i10}∼N\left(u_{i3},\sigma_{10}^{2}\right)$.We simulated gene expression data for 4800 genes, where $x_{ij}∼N\left(u_{i1},\sigma_{x}^{2}\right)$ for 1≤j≤1600, $x_{ij}∼N\left(u_{i2},\sigma_{x}^{2}\right)$ for 1601≤j≤3200, and $x_{ij}∼N\left(u_{i3},\sigma_{x}^{2}\right)$ for 3201≤j≤4800.

Similar to Simulations I and II, we set $N_{1}\in \left\{100,30\right\}$, $\sigma_{x}=0.5$, and $\sigma_{\mu }\in \left\{0,0.4,0.6\right\}$. In Simulation IIIA, we set $\sigma_{k}=2$ for 6≤k≤9, and $\sigma_{10}=1$, and in Simulation IIIB, we set $\sigma_{6}=0.01$, $\sigma_{10}=1$, and $\sigma_{k}=2$ for 6≤k≤9.

#### 3.1.2. Evaluation Benchmark

In Simulations I and II, the phenotypes were continuous, and we evaluated MANOVA, aSPU.ind, aSPU.ex, TATES, Fisher, minP, AFp, and AFz in terms of Type I error ($\sigma_{\mu }=0$) and power ($\sigma_{\mu }=0.4,0.6$). We also evaluated AFp and AFz for their accuracy of weight estimation. The Type I error and power were calculated by $\frac{\sum_{s=1}^{S}\sum_{j=1}^{p}I\left\{p_{j}^{\left(s\right)}<0.05\right\}}{p\times S}$, where S=500 is the number of simulated data for each setting, $p_{j}^{\left(s\right)}$ is the *p*-value of the *j*-th gene and the *s*-th simulated data of a generic method discussed in this paper, and I{.} is the indicator function. For the accuracy of the weight estimation of AFp and AFz, sensitivity $\frac{\sum_{s=1}^{S}\sum_{j=1}^{G}\sum_{k=1}^{K}{\hat{w}}_{jk}^{\left(s\right)}I\left\{w_{jk}=1\right\}}{\sum_{s=1}^{S}\sum_{j=1}^{p}\sum_{k=1}^{K}I\left\{w_{jk}=1\right\}}$(the proportion of weights estimated to be 1 when the truth is 1) and specificity $\frac{\sum_{s=1}^{S}\sum_{j=1}^{p}\sum_{k=1}^{K}\left(1-{\hat{w}}_{jk}^{\left(s\right)}\right)I\left\{w_{jk}=0\right\}}{\sum_{s=1}^{S}\sum_{j=1}^{p}\sum_{k=1}^{K}I\left\{w_{jk}=0\right\}}$ (the proportion of weights estimated to be 0 when the truth is 0) were used for evaluation. We also included average weight estimates for each phenotype and genes from 1 to 1600, 1601 to 3200, and 3201 to 4800 for further inspection (see Section 3.1.3 and Appendix A for details).

In Simulation III, the phenotypes were a mixture of count and continuous data, and MANOVA, aSPU.ind, and aSPU.ex are inapplicable. Therefore, we only evaluated TATES, Fisher, minP, AFp, and AFz in Simulation III. The benchmark criteria in Simulation III are the same as those in Simulations I and II.

#### 3.1.3. Simulation Results

Table 1 shows the Type I error, power, sensitivity, and specificity results of Simulation I with $N_{1}=100$. The simulation results of $N_{1}=30$ are shown in Appendix A with a similar pattern. All methods control type I error well, and the AFp and AFz methods generally perform among the best in terms of power. For example, in Simulation IA, all the phenotype–gene associations have similar effect sizes, and AFp (0.91) and AFz (0.9) have higher power than Fisher (0.87) and MANOVA (0.85) when $\sigma_{\mu }=0.6$. In Simulation IB, AFp and AFz, respectively, have powers of 0.96 and 0.97 when $\sigma_{\mu }=0.6$, which is comparable with minP (0.97) and higher than Fisher (0.89). In terms of 0–1 weight estimation, AFp has better sensitivity than AFz, and the gap is more significant in Simulation IB (0.49 and 0.78 for AFp compared with 0.29 and 0.31 for AFz). To dig further, in Appendix A, we calculate the average weight estimate for AFp and AFz for each phenotype and 1600 genes in Simulation I. In $\sigma_{\mu }=0.6$ and N=100, AFz selects phenotype 1 over phenotypes 2, 3, and 4 with a significantly higher proportion of 0.78 for genes 1∼1600, while AFp also evenly selects phenotypes 2, 3 and 4 (with proportions 0.72, 0.71, and 0.72). For genes 1601∼3200, AFz also selects phenotype 5 with a significantly higher proportion of 0.76 (proportions of phenotypes 6∼9 are 0.29, 0.28, 0.29, and 0.30), while AFp selects the phenotype 6∼9 with probabilities of 0.68, 0.68, 0.67 and 0.67, which are much closer to the probability of phenotype 5 (0.84). This means that when a gene has different effect sizes of associations with several phenotypes, AFz will only detect the association of the phenotype with the strongest association, while AFp can detect all associated phenotypes more evenly.

Appendix A show the Type I error, power, sensitivity, and specificity results of Simulation II for $N_{1}=100and30$, respectively. MANOVA cannot control Type I error well when there are confounders, while all other methods can control Type I error well. Similar to Simulation I, AFp and AFz generally perform among the best in terms of power, and AFp has better sensitivity than AFz in phenotype selection, especially when gene–phenotype association effect sizes are imbalanced (Appendix A).

Table 2 summarizes the results of Simulation III with $N_{1}=100$ when the phenotypes have count, binary, and continuous data. The simulation results of $N_{1}=30$ are shown in Appendix A with a similar pattern. In terms of power, AFp and AFz outperform the other three methods. For example, in Simulation IIIA with $N_{1}=100$, the power of TATES, minP, Fisher, AFz, and AFp are {0.43, 0.46, 0.51, 0.53, and 0.53} and {0.82, 0.79, 0.79, 0.9, and 0.91} for $\sigma_{\mu }=0.4$ and $\sigma_{\mu }=0.6$, respectively. Similar to Simulations I and II, AFp has much better sensitivity in terms of phenotype selection than AFz, and AFz only selects the phenotype with the strongest association (Appendix A) in most cases.

### 3.2. Real Application

#### 3.2.1. Application to Complex Lung Diseases

We applied MANOVA, aSPU.ind, aSPU.ex, TATES, Fisher, minP, AFp and AFz to a lung disease transcriptomic dataset with 319 patients, where the majority of patients were diagnosed as the two most representative lung disease subtypes: chronic obstructive pulmonary disease (COPD) and interstitial lung disease (ILD). Gene expression data were collected from Gene Expression Omnibus (GEO) GSE47460, and clinical information was obtained from the Lung Genomics Research Consortium (https://topmed.nhlbi.nih.gov/group/ltrc (accessed on 11 November 2019)). In this paper, FEV1, FVC, ratiopre, WBCDIFF1, and WBCDIFF4 were the five continuous phenotypes of interest. FEV1 (forced expiratory volume in 1 s) is the volume of air that can be forcibly blown out in the first 1 s after full inspiration. FVC (forced vital capacity) is the volume of air that can be forcibly blown out after full inspiration. Ratiopre is the ratio of FEV1 to FVC, and WBCDIFF1 and WBCDIFF4 are differential neutrophils (%) and differential eosinophils (%) in the white blood cells, respectively. Age, gender, and BMI were included as confounding covariates *X* in Equation (Equation 1) to calculate the input *p*-values for Fisher, minP, AFp, and AFz. After filtering samples with missing covariates, the final preprocessed dataset contained N=279 samples and p= 15,966 genes. We first evaluated MANOVA, aSPU.ind, aSPU.ex, TATES, Fisher, minP, AFp, and AFz by the numbers of significant genes detected by each method, followed by gene module identification through the AFp- and AFz-based methods.

Appendix A shows violin plots of $-\log_{10}$ (*p*-value) of all methods, where the significant genes are determined by Bonferroni correction with a cutoff of 0.05. We find that aSPU.ex has the most number of signficant genes (6092), followed by AFp (4367) and MANOVA (3973). We then focused on the significant genes of AFp- and AFz-based methods and categorized genes into gene modules. Table 3 shows the percentage of selected phenotypes for the 4367 and 3287 significant genes identified by AFp and AFz, respectively. Consistent with the simulation results, AFp has a more balanced distribution of phenotype selection, while AFz almost only selects ratiopre and unselects all the other phenotypes. For example, the percentages of genes selecting FEV1, FVC, and WBCDIFF1 are 79%, 48%, and 29% for AFp, while these numbers are 14%, 8%, and 1% for AFz. Appendix A shows the boxplot of the $-\log_{10}$ (*p*-value) of each phenotype for significant genes detected by both AFp and AFz methods, and it clearly indicates that the *p*-value of ratiopre is, on average, much smaller than that of other phenotypes. The AFz method almost only selects ratiopre and ignores others, which is consistent with the findings in Simulation IB, IIB, and IIIB (Appendix A). As a result, we only performed gene categorization on the AFp results, as below.

Following Section 2.6, we calculated the co-membership matrix of 4367 significant genes from AFp and utilized a tight clustering algorithm to cluster genes. A total of 1106 genes were clustered into seven clusters (C1, C2 ⋯ C7), where C1, C2, and C3 were closer to one another compared with other clusters, and they were categorized as module 1 (Figure 1a). Similarly, C6 and C7 were combined as module 4. Figure 1b shows the weight estimation. It again indicates that C1, C2, and C3 have similar patterns (up-regulated FEV1 and ratiopre and no association with FVC and WBCDIFF1), and C6 and C7 have similar patterns (down-regulated FEV1, ratiopre, and WBCDIFF4 and up-regulated FVC and WBCDIFF1), which is also confirmed by Appendix A, the heatmap of directed $-\log_{10}$ (*p*-values) of 1106 genes selected by the tight clustering method. To verify the weight estimations of the significant genes, we also calculated Spearman correlations between gene expressions and phenotypes in each cluster (Appendix A), which matched the pattern in Figure 1b well. In addition, we found that phenotype ratiopre has relatively strong associations with genes in all four gene modules (Appendix A) and is always assigned by 1 or −1 in weight estimation with high confidence (i.e., low variability index; see Appendix A). This observation justifies the common use of the FEV1/FVC ratio (ratiopre) in diagnosing obstructive and restrictive lung disease in current clinical practice [27,28].

We next conducted pathway enrichment analysis using Fisher’s exact test based on the Gene Ontology (GO), KEGG, and Reactome pathway databases to assess the biological relevance of genes and show the top 10 significant pathways for each module (Table 4). The top pathways for different modules depict distinct aspects of lung diseases. The top pathways in module 1 involve many DNA damage [29,30] and amino acid alternation/degradation pathways [31,32], which are known to be related to COPD in the literature. This set of genes is positively associated with FEV1 and ratiopre (and some with WBCDIFF4). Module 2 is enriched in many immune response pathways. The immune system needs to react promptly and adequately to potential challenges posed by microbes and particles, while at the same time avoiding extensive tissue damage. Many studies have shown the association between immune response and lung diseases, such as Toll-like receptor and NOD-like receptor [33,34] and kinase-based protein signaling cascades [35]. This set of genes is negatively associated with FEV1, ratiopre, and WBCDIFF4 and positively associated with WBCDIFF1. Module 3 clearly indicates many extracellular matrix (ECM) structure pathways, which provide structural support and stability to the lung. Changes in the ECM in the airway or parenchymal tissues are now recognized in the pathological profiles of many respiratory diseases, including COPD [36]. This gene module is positively associated with FEV1, ratiopre, and WBCDIFF4 but negatively associated with FVC (and some to WBCDIFF1). The top pathways in module 4 include pathways related to cancer and vasculature development. COPD is a risk factor for lung cancer as they have many shared driving factors and genetic effects [37]. Additionally, COPD is a risk factor for major cancers developing outside of the lung, including bladder cancer and pancreatic cancer [38,39]. Furthermore, angiogenesis (vasculature development) is a shared phenomenon for both cancer and COPD [40], which may indicate the molecular connection between COPD and cancers. Genes in this module are positively correlated with FVC, with some being positively correlated with WBCDIFF1 but negatively correlated with FEV1, ratiopre, and WBCDIFF4.

#### 3.2.2. Application to Breast Cancer

We applied Fisher, minP, AFp, and AFz to a breast cancer transcriptomic dataset with 1981 patients collected from the Molecular Taxonomy of Breast Cancer International Consortium (METABRIC, https://www.cbioportal.org/study/summary?id=brca_metabric (accessed on 20 February 2023)). We considered tumor grade (binary), lymph node status (count), overall survival (survival), and tumor size (continuous) as the four phenotypes of interest, with age as the confounding covariate. We performed a pre-processing step that kept the 10,000 genes with the largest coefficients of variation. The purpose of this was to remove housekeeping genes. Patients with missing outcomes or confounders were further removed, and 1708 samples were left. As we considered a mixture of phenotypes of different data types, only Fisher, minP, AFp, and AFz can be applied.

Appendix A shows violin plots of $-\log_{10}$ (*p*-values) of the four methods, where the significant genes are determined by Bonferroni correction with a cutoff of 0.05. We found that AFp has the most significant genes (5856), followed by EW (5194) and AFz (4590). Similar to Section 3.2.1, we then focused on the significant genes of the AFp-based method and categorized these genes into gene modules.

Following Section 2.6, we calculated a co-membership matrix of 5856 significant genes from AFp and utilized the tight clustering algorithm to cluster genes. A total of 3560 genes were clustered into five gene modules (clusters) (Figure 2a). Figure 2b shows the weight estimation, which is verified by Appendix A. Specifically, we find up-regulation for all phenotypes in gene module 1 (tumor grade, lymph node status, overall survival, and tumor size) and down-regulation for all phenotypes in gene module 2. Gene modules 3 and 5 are down-regulated for lymph node status, grades, and survival, while gene module 4 is up-regulated for lymph node status and grade. Additionally, we find that tumor grade and lymph node status have relatively strong associations with genes when compared to overall survival and tumor size. Lastly, we also find that grade has highly strong associations with genes in all the five genes modules (Appendix A) and is always assigned by 1 or −1 in weight estimation with high confidence (Figure 2b and Appendix A).

We next conducted pathway enrichment analysis using Fisher’s exact test based on the Gene Ontology (GO), KEGG, and Reactome pathway databases to assess the biological relevance of genes and show the 10 most significant pathways for each module (Table 5). The top pathways for different modules depict distinct aspects of breast cancer. The top pathways in gene module 1 involve cell cycle processes; the top pathways in gene module 2 are related to metabolic processes; the top pathways in gene module 3 involve transport mechanisms; the top pathways in gene module 4 are closely related to the immune system; and the top pathways in gene module 5 relate to the Golgi apparatus. All five aspects play important roles in cancer [41,42,43,44,45,46].

#### 3.2.3. Application to Brain Aging

We lastly applied Fisher, minP, AFp, and AFz to a brain aging dataset from [47]. The dataset contains 210 samples’ transcriptomic profiles in the human prefrontal cortex (Brodmann’s area 11, https://www.ncbi.nlm.nih.gov/geo/query/acc.cgi?acc=GSE71620 (accessed on 20 February 2023)). We considered age (continuous), sex (binary), and recorded pH levels of the tissue sample (continuous) as the three phenotypes of interest, and the postmortem interval between the time of death and the time of tissue sample collection as well as the RNA integrity number for each tissue sample as confounders. Similar to the breast cancer dataset, we performed a pre-processing step that kept the 10,000 genes with the largest coefficients of variation to remove housekeeping genes. Since we considered a mixture of phenotypes of different data types, only Fisher, minP, AFp, and AFz can be applied.

Appendix A shows violin plots of $-\log_{10}$ (*p*-values) of the four methods, where the significant genes are determined by the Benjamini–Hochberg correction with a cutoff 0.05. We find that AFp has the most significant genes (385), followed by AFz (303) and minP (272). Similar to Section 3.2.1 and Section 3.2.2, we then focused on the significant genes of the AFp-based method and categorized genes into gene modules.

Following Section 2.6, we calculated a co-membership matrix of 385 significant genes from AFp and utilized a tight clustering algorithm to cluster genes. A total of 290 genes were clustered into three gene modules (clusters) (Appendix A). Appendix A shows the weight estimation, which is verified by Appendix A. Specifically, we find up-regulation of age in gene modules 1 and 2 and down-regulation of age in gene module 3. Sex is down-regulated by genes in the second module, and pH seems to be irrelevant for all three modules. Similar to Section 3.2.1 and Section 3.2.2, we next performed a pathway enrichment analysis using Fisher’s exact test based on the Gene Ontology (GO), KEGG, and Reactome pathway databases to assess the biological relevance of the genes and show the 10 most significant pathways for each module (Appendix A). The top pathways in gene module 1 regulate the innate immune system; the top pathways in gene module 2 are related to immune signaling; and the top pathways in gene module 3 involve cell signaling. All three categories of pathways from the gene modules connect to the brain aging process [48,49,50,51].

## 4. Discussion

In this paper, we modified two *p*-value combination methods, AFp and AFz, to multivariate phenotype–gene association studies. Compared with traditional methods targeting the UIT test between each gene and all phenotypes, AFp and AFz can efficiently combine heterogeneous effects in the input *p*-values. Both methods facilitate the selection of phenotypes associated with the gene of interest. A bootstrap algorithm and gene cluster approach identified gene modules with distinct phenotype association patterns. Pathway enrichment analysis for identified gene modules elucidated the disease mechanisms underlying multivariate phenotype–gene associations.

From extensive simulations and real application examples, we clearly showed that AFp and AFz have robust and competitive statistical power and Type 1 error control for accommodating an association analysis of heterogeneous phenotypes in complex diseases. Moreover, AFp has better sensitivity to phenotype selection compared with AFz, especially when heterogeneous association effect sizes exist across phenotypes. In conclusion, we recommend the AFp-based method for multivariate phenotype–gene association studies, for the subset identification of phenotypes associated with the gene, and downstream analyses, such as gene categorization and pathway enrichment analysis. Our R package to implement AFp, AFz, and all existing methods is available at https://github.com/hung-ching-chang/MultiPhenoAssoc, along with all data and source code used in this paper.

One limitation of our proposed method is the relatively heavy computational demand of permutation analysis, which is essential for taking the correlation structure of the phenotypes into consideration. To relieve the computational burden, we utilized the R package “Rfast” [52] to speed up and also optimize our code to put it in an affordable range for general omics applications. In the lung disease application (K=5, N=279, and p= 15,966), with 50 times bootstrapping using 50 computing threads, it takes approximately 2 h to implement the AFp-based method. Developing methods for further improving the computation is a future direction.

## Figures and Tables

**Figure 1 genes-14-00798-f001:**
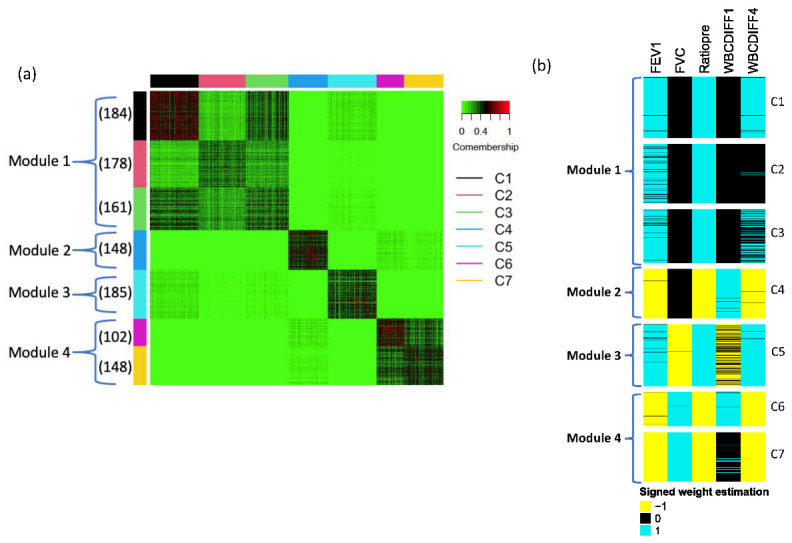
(**a**) The heatmap of comembership matrix of seven clusters identified in the complex lung diseases dataset. Red color means two genes are close. The number in the parentheses indicates the sample size of each cluster. (**b**) The weight estimation of each gene (blue represents 1, black represents 0 and yellow represents −1).

**Figure 2 genes-14-00798-f002:**
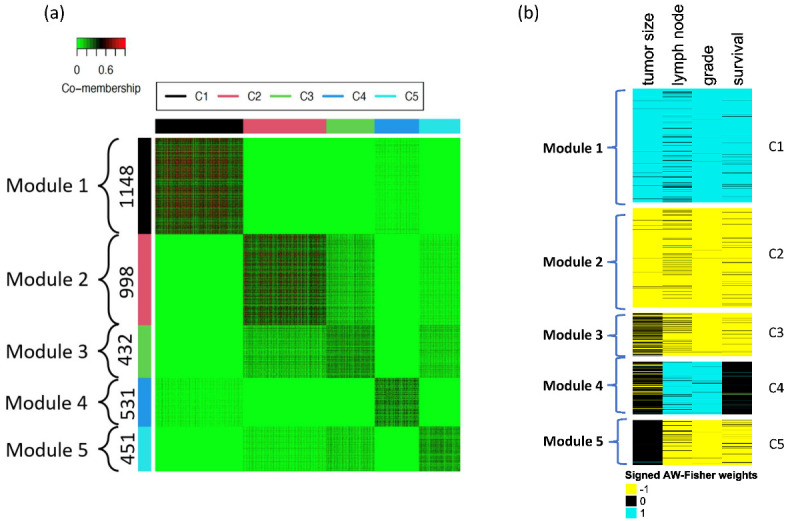
(**a**) The heatmap of the co-membership matrix of five clusters identified in the breast cancer (METABRIC) dataset. Red color means two genes are close. The number in the parentheses indicates the sample size of each cluster. (**b**) The weight estimation of each gene (blue represent 1, black represents 0, and yellow represents −1).

**Table 1 genes-14-00798-t001:** Results of Simulations IA and IB when $N_{1}=100$. For $\sigma_{\mu }=0$, the value indicates Type I error control, and for $\sigma_{\mu }=0.4$ and 0.6, the value indicates power.

Benchmark	Method	Simulation IA			Simulation IB		
		$\sigma_{\mu }$ = 0	$\sigma_{\mu }$ = 0.4	$\sigma_{\mu }$ = 0.6	$\sigma_{\mu }$ = 0	$\sigma_{\mu }$ = 0.4	$\sigma_{\mu }$ = 0.6
power & type I error	MANOVA	0.05	0.36	0.85	0.05	0.76	0.93
	aSPU.ind	0.05	0.42	0.67	0.05	0.46	0.69
	aSPU.ex	0.05	0.41	0.67	0.05	0.45	0.7
	TATES	0.05	0.25	0.79	0.05	0.74	0.96
	minP	0.05	0.33	0.85	0.05	0.77	0.97
	Fisher	0.05	0.44	0.87	0.05	0.71	0.89
	AFz	0.05	0.4	0.9	0.05	0.77	0.97
	AFp	0.05	0.42	0.91	0.05	0.77	0.96
Sensitivity	AFz	-	0.34	0.6	-	0.29	0.31
	AFp	-	0.38	0.72	-	0.49	0.78
Specificity	AFz	-	0.79	0.76	-	0.86	0.89
	AFp	-	0.77	0.68	-	0.74	0.67

**Table 2 genes-14-00798-t002:** Results of Simulations IIIA and IIIB when $N_{1}=100$. For $\sigma_{\mu }=0$, the value indicates Type I error control, and for $\sigma_{\mu }=0.4$ and 0.6, the value indicates power.

Benchmark	Method	Simulation IIIA			Simulation IIIB		
		$\sigma_{\mu }$ = 0	$\sigma_{\mu }$ = 0.4	$\sigma_{\mu }$ = 0.6	$\sigma_{\mu }$ = 0	$\sigma_{\mu }$ = 0.4	$\sigma_{\mu }$ = 0.6
Power & Ttype I error	TATES	0.05	0.43	0.82	0.05	0.74	0.96
	minP	0.05	0.46	0.79	0.05	0.76	0.95
	Fisher	0.05	0.51	0.79	0.05	0.7	0.84
	AFz	0.05	0.53	0.9	0.05	0.77	0.97
	AFp	0.05	0.53	0.91	0.05	0.77	0.96
Sensitivity	AFz	-	0.48	0.72	-	0.3	0.43
	AFp	-	0.55	0.81	-	0.6	0.85
Specificity	AFz	-	0.8	0.81	-	0.8	0.82
	AFp	-	0.76	0.68	-	0.73	0.66

**Table 3 genes-14-00798-t003:** The proportion of weight estimated to be 1 for significant genes (4367 and 3287 for AFp and AFz, respectively, determined by Bonferroni correction with cutoff 0.05) of AFp and AFz methods.

Method	FEV1	FVC	Ratiopre	WBCDIFF1	WBCDIFF4
AFp	79%	48%	99%	29%	67%
AFz	14%	8%	98%	1%	6%

**Table 4 genes-14-00798-t004:** The pathway enrichment analysis of each module by GO, KEGG, and Reactome pathway databases for the lung disease dataset. The * sign indicates the *p*-value is significant under a false discovery rate of 0.05.

Pathway	*p* Value
module 1	
GO:BP double-strand break repair	4.37 $\times 10^{-3}$
Reactome double-strand break repair	4.37 $\times 10^{-3}$
KEGG Valine, leucine and isoleucine degradation	6.78 $\times 10^{-3}$
Reactome Branched-chain amino acid catabolism	7.82 $\times 10^{-3}$
Reactome Homologous recombination repair of replication-independent double-strand breaks	1.42 $\times 10^{-2}$
GO:MF phosphotransferase activity, phosphate group as acceptor	1.98 $\times 10^{-2}$
GO:BP gamete generation	2.67 $\times 10^{-2}$
GO:BP sexual reproduction	2.68 $\times 10^{-2}$
GO:MF motor activity	3.39 $\times 10^{-2}$
GO:MF nucleobase-containing compound kinase activity	3.39 $\times 10^{-2}$
module 2	
KEGG Toll-like receptor signaling pathway	8.47 $\times 10^{-6}$ *
KEGG NOD-like receptor signaling pathway	1.03 $\times 10^{-5}$ *
KEGG MAPK signaling pathway	3.91 $\times 10^{-5}$ *
GO:BP response to stress	3.95 $\times 10^{-5}$ *
KEGG cytosolic DNA-sensing pathway	4.13 $\times 10^{-5}$ *
GO:MF enzyme binding	7.46 $\times 10^{-5}$ *
GO:BP protein kinase cascade	8.41 $\times 10^{-5}$ *
GO:MF rho gtpase activator activity	8.93 $\times 10^{-5}$ *
Reactome NFkB and MAP kinase activation mediated by TLR4 signaling repertoire	1.28 $\times 10^{-4}$ *
GO:BP regulation of protein kinase activity	1.55 $\times 10^{-4}$ *
module 3	
Reactome extracellular matrix organization	1.48 $\times 10^{-8}$ *
Reactome collagen formation	1.64 $\times 10^{-6}$ *
GO:CC proteinaceous extracellular matrix	3.28 $\times 10^{-6}$ *
GO:CC extracellular matrix	3.97 $\times 10^{-6}$ *
GO:CC extracellular region part	2.66 $\times 10^{-5}$ *
GO:CC collagen trimer	7.31 $\times 10^{-5}$ *
GO:CC extracellular region	9.26 $\times 10^{-5}$ *
GO:CC extracellular matrix component	1.33 $\times 10^{-4}$ *
GO:MF glycosaminoglycan binding	3.30 $\times 10^{-4}$
Reactome diabetes pathways	3.63 $\times 10^{-4}$
module 4	
KEGG MAPK signaling pathway	1.87 $\times 10^{-4}$
KEGG dorso-ventral axis formation	1.89 $\times 10^{-4}$
KEGG bladder cancer	2.62 $\times 10^{-4}$
KEGG pancreatic cancer	2.84 $\times 10^{-4}$
GO:MF neurotransmitter binding	3.80 $\times 10^{-4}$
GO:BP angiogenesis	4.28 $\times 10^{-4}$
KEGG pathways in cancer	4.55 $\times 10^{-4}$
GO:BP organ development	5.51 $\times 10^{-4}$
GO:BP vasculature development	7.34 $\times 10^{-4}$
GO:BP anatomical structure formation involved in morphogenesis	9.83 $\times 10^{-4}$

**Table 5 genes-14-00798-t005:** The pathway enrichment analysis of each module by GO, KEGG, and Reactome pathway databases for the breast cancer (METABRIC) dataset. The * sign indicates the *p*-value is significant under a false discovery rate of 0.05.

Pathway	*p* Value
Module 1 (cell cycle)	
Reactome cell cycle, mitotic	1.63 $\times 10^{-45}$ *
Reactome cell cycle	1.73 $\times 10^{-45}$ *
Reactome DNA replication	1.90 $\times 10^{-38}$ *
Reactome, itotic M-M/G1 phases	1.03 $\times 10^{-32}$ *
GO:BP cell cycle process	7.73 $\times 10^{-22}$ *
GO:BP cell cycle	4.82 $\times 10^{-21}$ *
GO:BP mitotic cell cycle	6.48 $\times 10^{-21}$ *
Reactome mitotic prometaphase	3.99 $\times 10^{-20}$ *
GO:BP cell cycle phase	5.42 $\times 10^{-19}$ *
GO:BP mitotic M phase	3.02 $\times 10^{-18}$ *
Module 2 (metabolic processes)	
Reactome Synthesis of bile acids and bile salts	2.89 $\times 10^{-4}$ *
Reactome Nuclear signaling by ERBB4	8.53 $\times 10^{-4}$ *
Reactome Bile acid and bile salt metabolism	1.44 $\times 10^{-3}$ *
KEGG primary bile acid biosynthesis	1.60 $\times 10^{-3}$ *
Reactome peroxisomal lipid metabolism	4.58 $\times 10^{-3}$ *
GO:BP carboxylic acid metabolic process	5.82 $\times 10^{-3}$ *
Reactome G alpha (s) signalling events	6.01 $\times 10^{-3}$ *
GO:BP organic acid metabolic process	7.04 $\times 10^{-3}$ *
GO:BP sodium ion transport	8.23 $\times 10^{-3}$ *
GO:BP regulation of cytoskeleton organization	8.44 $\times 10^{-3}$ *
Module 3 (transport mechanisms)	
Reactome ABCA transporters in lipid homeostasis	8.90 $\times 10^{-4}$ *
GO:CC integral component of organelle membrane	4.67 $\times 10^{-3}$ *
GO:CC intrinsic component of organelle membrane	6.10 $\times 10^{-3}$ *
GO:BP secretion by cell	7.25 $\times 10^{-3}$ *
Reactome PKB-mediated events	7.94 $\times 10^{-3}$ *
KEGG ether lipid metabolism	7.94 $\times 10^{-3}$ *
Reactome acyl chain remodelling of PE	1.01 $\times 10^{-2}$ *
GO:BP secretion	1.02 $\times 10^{-2}$ *
GO:BP synaptic transmission	1.20 $\times 10^{-2}$ *
GO:BP secretory pathway	1.22 $\times 10^{-2}$ *
Module 4 (immune system)	
GO:BP immune system process	2.34 $\times 10^{-36}$ *
GO:BP immune response	6.54 $\times 10^{-29}$ *
Reactome adaptive immune system	6.21 $\times 10^{-25}$ *
Reactome immunoregulatory interactions between a lymphoid and a non-lymphoid cell	3.60 $\times 10^{-22}$ *
KEGG natural killer cell-mediated cytotoxicity	7.20 $\times 10^{-16}$ *
KEGG primary immunodeficiency	4.20 $\times 10^{-15}$ *
GO:BP defense response	9.49 $\times 10^{-15}$ *
KEGG T cell receptor signaling pathway	3.84 $\times 10^{-11}$ *
GO:BP T cell activation	6.67 $\times 10^{-11}$ *
GO:BP regulation of immune system process	1.96 $\times 10^{-10}$ *
Module 5 (Golgi apparatus)	
Reactome generic transcription pathway	1.35 $\times 10^{-3}$ *
GO:CC Golgi apparatus part	1.53 $\times 10^{-3}$ *
GO:CC coated vesicle	4.58 $\times 10^{-3}$ *
GO:CC Golgi-associated vesicle	6.08 $\times 10^{-3}$ *
GO:CC Golgi apparatus	1.59 $\times 10^{-2}$ *
Reactome xenobiotics	1.85 $\times 10^{-2}$ *
GO:BP Golgi vesicle transport	2.43 $\times 10^{-2}$ *
Reactome Pre-NOTCH Transcription and Translation	2.76 $\times 10^{-2}$ *
Reactome Pre-NOTCH Expression and Processing	3.14 $\times 10^{-2}$ *
GO:BP adenylate cyclase-activating G-protein-coupled receptor signaling pathway	4.74 $\times 10^{-2}$ *

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
