# Peer review of "Adaptively Integrative Association between Multivariate Phenotypes and Transcriptomic Data for Complex Diseases"

_genes, 2023, doi:10.3390/genes14040798_

Round 1

Reviewer 1 Report

1.     The authors acknowledge the limitation that their method is computationally intensive. However, their simulation results (tables 1 and 2) show a very little or no improvement over the other methods which are much easier to understand and less computationally expensive. It doesn’t seem reasonable to me to apply a such computationally heavier method for a tiny bit of improvement in power given that they have all type I errors controlled.

2.     It is very surprising to me that the authors chose only 150 genes in their simulations while, in reality, these datasets involve thousands of genes. Even in their real data application, they had more than 15000 genes. It is necessary to check the performances of their methods with a realistic number of genes. The sample size considered in the simulation is 100. I understand that their real data contains more samples. But I would like to see how the methods perform with much smaller sample sizes, such as 20 or 30, which is generally the case in most genomic studies.

3.     I am a bit confused about the aim of their method. Based on the null and alternative hypotheses in page 2, are we interested in identifying the genes that are associated with at least one of the phenotypes? If it is so, doesn’t it make more sense to identify genes that are associated with either all or most of the phenotypes?

4.     It would be really helpful if the authors can provide a justification of the AFp and AFz methods selecting the minimum p-value/score, in this paper.

5.     The statistic given in page 4, section 2.2, which is the weighted sum of log-transformed p-values, considers weights as either 0 or 1 depending on whether the gene is associated with a phenotype. That means, we are just combining p-values that are significant. How is the significance decided in this step? Do we need to choose a cutoff (e.g., 0.05) to consider a gene significant here? If so, are we not already deciding if a gene is significantly associated with a phenotype?

The authors state that their weight vector can be used to determine which specific phenotypes a gene is associated with. But can't we just determine that from the initial p-values obtained from the phenotype-gene association tests?

6.     In section 2.6, it is not clear to me how clustering the genes will help in reducing the no. of phenotype association pattern types. An explanation would be helpful for general readers.

7.     I appreciate that the authors considered a few variations in their simulations with continuous and count phenotypes. Another interesting scenario would be to consider categorical phenotypes.

8.     Finally, the authors selected a Bonferroni threshold of 0.05. The Bonferroni correction method is a very conservation multiplicity correction method but still it yielded quite a lot significant number of genes. Does it mean there is a possibility of false discoveries (inflated type I error) in the real data?

Author Response

Please see attached response letter.

Reviewer 2 Report

The authors of this paper developed a method for finding differential expression genes in medical data with multivariate phenotypes based on the conjunction null hypothesis.

commants:

1. Please test your method on two more RNA or microarray data sets. It is not correct to draw conclusions based on the results of only one dataset to demonstrate that your method is the best method.

2. Please compare your method to the DEGS chosen by Random Forests on medical data with bivariate traits.

3. please show the prediction ability of selected top genes in  major module with roc curves.

Author Response

Please see attached response letter.
